# Meta-Analysis of the Effects of Biochar Application on the Diversity of Soil Bacteria and Fungi

**DOI:** 10.3390/microorganisms11030641

**Published:** 2023-03-02

**Authors:** Mingyu Wang, Xiaoying Yu, Xiaohong Weng, Xiannan Zeng, Mengsha Li, Xin Sui

**Affiliations:** 1Engineering Research Center of Agricultural Microbiology Technology, Ministry of Education & Heilongjiang Provincial Key Laboratory of Ecological Restoration and Resource Utilization for Cold Region & Key Laboratory of Microbiology, College of Heilongjiang Province & School of Life Sciences, Heilongjiang University, Harbin 150080, China; 2Institute of Crop Cultivation and Tillage, Heilongjiang Academy of Agricultural Sciences, Harbin 150088, China; 3Institute of Nature and Ecology, Heilongjiang Academy of Sciences, Harbin 150040, China

**Keywords:** alpha diversity, bacteria, biochar, soil microbial diversity, meta-analysis

## Abstract

Biochar is increasingly being used for soil improvement, but the effects on microbial diversity in soil are still ambiguous due to contrasting results reported in the literature. We conducted a meta-analysis to clarify the effect of biochar addition on soil bacterial and fungal diversity with an increase in Shannon or Chao1 index as the outcome. Different experimental setups, quantitative levels of biochar addition, various biochar source materials and preparation temperatures, and the effect of natural precipitation in field experiments were the investigated variables. From a total of 95 publications identified for analysis, 384 datasets for Shannon index and 277 datasets for Chao1 index were extracted that described the bacterial diversity in the soils, of which field experiments and locations in China dominated. The application of biochar in soil significantly increased the diversity of soil bacteria but it had no significant effect on the diversity of fungi. Of the different experimental setups, the largest increase in bacterial diversity was seen for field experiments, followed by pot experiments, but laboratory and greenhouse settings did not report a significant increase. In field experiments, natural precipitation had a strong effect, and biochar increased bacterial diversity most in humid conditions (mean annual precipitation, MAP > 800 mm), followed by semi-arid conditions (MAP 200–400 mm). Biochar prepared from herbaceous materials was more effective to increase bacterial diversity than other raw materials and the optimal pyrolysis temperature was 350–550 °C. Addition of biochar at various levels produced inconclusive data for Chao1 and Shannon indices, and its effect was less strong than that of the other assessed variables.

## 1. Introduction

Biochar is a mixture of organic materials that is typically obtained via pyrolysis of waste biomass under low-oxygen conditions [1,2,3]. Its addition to soil is an effective way to enhance soil quality and productivity [4,5,6]. This reuse of what would otherwise be agricultural waste has become an emerging technology for sustainable soil management to add biomass as organic amendment [7]. Biochar can enhance terrestrial carbon sequestration and provides a tool for mitigation of greenhouse gases [8,9,10]. Its application can improve soil fertility and plant productivity [11,12,13], as well as improving soil porosity [14,15,16]. Compared to its effect on soil characteristics and fertility, the effects of biochar on the microbial communities of soil have been less thoroughly assessed [17,18].

Microorganisms in the soil can directly or indirectly participate in soil activities [19,20,21,22]. It has been demonstrated by several studies that biochar can increase soil microbial diversity [23,24,25,26,27]. However, the mechanism by which biochar affects soil microbial diversity is less clear. Three possible mechanisms have been proposed in the literature. (i) The relatively large surface area and porosity of biochar can provide a habitat for soil microorganisms, providing a living space while protecting them from predation and exposure to harmful conditions [28,29]. (ii) The altered physical properties of the soil by addition of biochar may support growth and activity of soil microorganisms, for example, by improving the soil cation exchange capacity (CEC), soil pore space and soil water content, which may affect the soil microbial diversity positively [30,31,32]. (iii) By adding biochar, the growth and reproduction of soil microorganisms would be supported [33] as addition of biochar alters the pH and adds organic carbon, nitrogen phosphorus and potassium of the soil [34,35] and these nutrients will increase soil microbial diversity [36,37]. Added nutrients may promote nutrient cycling in the soil and thus enhance soil microbial diversity.

Soil microorganisms are the main decomposers in terrestrial ecosystems and are very active members of the soil, with a wide variety of species [38], of which only a limited number have so far been characterized. Soil microorganisms are a vital part of the biogeochemical cycle, actively contributing to material cycling and energy flow, such as the decomposition of organic matter, nutrient cycling and the biodegradation of organic pollutants [39,40,41,42,43]. Soil microorganisms play a role in maintaining the stability of the structure and function of the soil ecosystem [44,45]. They can be closely associated with plants in a rhizosphere where they participate in the transformation of soil organic matter to provide nutrients to plants, and have an important influence on the structure of plant communities [46]. Based on these properties, soil microorganisms can be used to assess the quality and health of a given soil [47,48], and a healthy soil microbial community, characterized by a large diversity, is a primary prerequisite for good soil quality and stable soil ecosystem structure and function [49,50,51].

The potential of biochar application on CO_2_ flux is still under debate [52], as the direction and magnitude of effects seem to depend on soil properties, land-use type, experimental methods, vegetation presence, or biochar characteristics that vary with feedstock type and pyrolysis conditions. The same probably applies to the effect on microbial diversity. Laboratory and field studies have been conducted to study the biochar application to different soils and most of these reported positive effects on the diversity of bacteria and fungi in the soil [53,54,55,56,57], but other studies found that the application of biochar had no significant effect on soil microbial diversity [58,59]. This warrants a systematic comparison of the data available from the literature.

To address this, we performed a meta-analysis, based on the concept that was first proposed by Glass in 1976 [60]. A meta-analysis is highly suitable for a comprehensive evaluation and quantitative study of existing research results [61,62]. From a total of 95 publications identified for analysis, 384 datasets for Shannon index and 277 datasets for Chao1 index were extracted that described the bacterial diversity in the soils, of which field experiments and locations in China dominated. These were synthesized to examine the responses of soil microbial diversity to biochar applications, with the reported Shannon and Chao1 index as the outcome. This allowed a quantitative examination of the effect size of biochar application on soil microbial diversity, and enabled the identification of the key factors that influence the response of microbial diversity.

## 2. Materials and Methods

### 2.1. Data Collection and Extraction

In order to systematically clarify the effect of biochar addition on soil microbial diversity, we conducted a meta-analysis of the published literature on this subject. The data for the meta-analysis were obtained from three literature databases: Google Scholar (http://scholar.google.com/), Web of Sciences (WoS) (http://apps.webofknowledge.com/) and China National Knowledge Infrastructure (CNKI, http://www.cnki.net/). The search terms were “biochar” combined with “soil microorganism”, “soil microbial diversity”, “Shannon index” or “Chao1 index”, and the data collection was completed in June 2022.

The following inclusion criteria were used to screen the search hits: (i) studies had to include replicated biochar treatments (of at least one biochar addition) and controls (no biochar); (ii) each treatment must include three or more repetitions; (iii) at least one microbial diversity metric (Shannon index or Chao1 index) had to be reported. Publications in languages other than English or Chinese were excluded Since this study focused on the effect of biochar addition on soil microbial diversity, we collected target and explanatory variables from the included literature. For soil microbial diversity as a target variable, we collected the reported Shannon index and Chao1 index of bacteria and fungi. The sample size (n) for the biochar addition and the value of mean, standard deviation (SD) of the reported diversity index and the control treatment were extracted. If the SD was not indicated, it was calculated from the SE as SD = SE√N. In cases where neither SD nor SE was reported, the SD was calculated from the mean [63]. When the data were presented in graphical form only, quantitative data were extracted by Web Plot Digitizer software4.6 [64]. As explanatory variables we extracted the setup of the experiments and the levels of biochar application (see below), the raw material from which the biochar was produced, the biochar preparation temperature, and, in case of field studies, the natural precipitation level. Studies containing multiple datasets for these variables were treated as multiple independent studies.

A total of 95 papers were identified from WOS, Google Scholar and CNKI database that met the criteria. From those that represented field studies, we collated the location of the sampling sites and plotted their latitude and longitude data on a map (Figure 1).

### 2.2. Data Grouping

We grouped the various explanatory variables as follows. The experiments were grouped into field, greenhouse, pot, and laboratory experiments. Data from field studies were grouped for MAP with arid (<200 mm), semi-arid (200–400 mm MAP), semi-humid (400–800 mm) and humid conditions (≥800 mm). After standardizing the units of biochar addition (with 27 t·hm^−2^ = 1% and 81 t·hm^−2^ = 3%) [65], the biochar addition was grouped into three levels of low (<1.5%), medium (1.5–3%), and high (>3%) application rates. The biochar feedstocks were grouped based on their source type as previously described [66], with manure, wood, herbaceous materials and lignocellulosic wastes, to which we added the group of domestic waste (biochar made from laboratory wastewater sludge, municipal sludge, household waste or discarded mushroom substrate) and modified biochar (iron-based, calcium-based, or manganese-based modified biochar). The pyrolytic temperatures used for biochar preparation were grouped as low (≤350 °C), medium (350–500 °C) and high (≥500 °C).

### 2.3. Calculation of Effect Size and Variance

The method for calculating the effect size and variance was taken from the literature [67]. The natural log-transformed response ratio (*LRR*) was used to measure the effect size and was calculated from the mean treatment value *Xt* and the mean value in the control *Xc* as follows:$$LRR=\ln\left(Xt/Xc\right)$$

The variance (*v*) of each individual effect size was calculated as follows: $$v=\frac{St^{2}}{n_{t}\bar{Xt^{2}}}+\frac{Sc^{2}}{n_{c}\bar{Xc^{2}}}$$
where *n_t_* represents the number of the treatment samples, *n_c_* the number of the control samples, *St* is the standard deviation (SD) of the treatment and *Sc* that of the control groups. We filled in any missing SD values in the collated data by using the “impute_SD” function in the “metagear” package of R4.1.3 software [68]. 

The weighted response ratio (*LRR*_++)_, its standard error *S*(*LRR*_++_) and the 95% confidence interval (95%Cl) were calculated as follows:$$LRR_{++}=\frac{\sum \frac{m}{i=1}\sum \frac{ki}{j=1}W_{i}{}_{j}RR_{i}{}_{j}}{\sum \frac{m}{i=1}\sum \frac{ki}{j=1}W_{i}{}_{j}}$$
$$S\left(LRR_{++}\right)=\sqrt{\frac{1}{\sum \frac{m}{i=1}\sum \frac{ki}{j=1}W_{ij}}}$$
$$95\%\mathrm{Cl}=LRR_{++}\pm 1.96•S\left(LRR_{++}\right)$$

Positive and negative values of *LRR*_++_ represent positive and negative effects, respectively. If the 95% confidence interval (Cl_95%_) reached 0, the biochar applications was considered to have no significant effect on the assessed variables. A Cl_95%_ above 0 (upper and lower bound) indicated the biochar applications significantly increased the soil microbial diversity (*p* < 0.05) whereas a Cl_95%_ below zero indicated a biochar application significantly decreased the soil microbial diversity (*p* < 0.05).

For each observation, we calculated the effect size as described in the literature [67] using the “escalc” function in the R package “metafor” [69]. A random effects model was first used to assess the overall responses, after which a mixed-effects meta-regression analysis was used that included all fixed factors for each response variable. In the latter model, we used the restricted maximum likelihood (REML) [70] method to estimate the between-case variance. In the random effects model, we used the restricted maximum likelihood method to calculate *tau*^2^, the inter-case variance (the difference in effect value due to different cases) [71]. This, and vi as the intra-case variance was used to calculate the weighing factor (*W_ij_*):$$W_{\mathrm{ij}}=\frac{1}{vi+tau^{2}}$$

As described in the literature [67], we evaluated the heterogeneity of effect sizes with the *Qt*-test to determine whether the models could explain a significant amount of variation. For this, *Qm* (a Wald-type test of model coefficients and *Qe*, which indicates the residual heterogeneity of unknown factors were summed a significant *Qm*-statistic indicates that the moderators contribute to the heterogeneity in effect sizes [72].
Qt=Qm+Qe

This determined the overall heterogeneity and would indicate a need to introduce different moderators to explain the observations [73]. We used the “~factor-1” command in the “metafor” package to calculate direct estimates of the cumulative effect of different moderators [74]. For a meta-analysis it is recommended that the results are reported truthfully and without bias, regardless of statistical significance [75]. Common methods used to assess publication bias are funnel plots and Egger (regression) tests, which result in funnel asymmetry when publication bias is present [76]. To test for publication bias in our data, we used funnel plots, Egger’s test [77], and fail-safe numbers [78]. The “Trim and fill” algorithm was also used to identify and adjust for funnel asymmetry from publication standard error [79].

### 2.4. Graphical Repreparation of the Results

Caterpillar plots were prepared to show effect sizes calculated for each study with the “ggplot2” package [80] in R software. Orchard plots were prepared with the “Orchard” package [81] in R software to show the mean estimate of soil microbial diversity index, confidence intervals, prediction intervals and individual effect sizes and their precision (inverse variance) for different types of experimental setups. Forest plots with subgroups were prepared to calculate estimates of the cumulative effect of different moderators with the “forestplot” package in R software. In all plots, the number of datasets for any given group is specified as K.

## 3. Results

### 3.1. Effects of Biochar Addition on Diversity Indices of Soil Bacterial Communities

From a total of 95 publications identified for analysis, 384 datasets for Shannon index and 277 datasets for Chao1 index were extracted that described the bacterial diversity in the soils.

The application of biochar significantly increased the bacterial Shannon index compared to the control group (Figure 2, *p* < 0.05). The overall weighted mean effect size estimate was slightly above zero, but the overall heterogeneity was large, with *Qt* = 34,801.5 (not shown in the figure), requiring the introduction of explanatory variables.

#### 3.1.1. Effects of Individual Parameters on the Soil Bacterial Diversity Shannon Index 

Considering subgroups for the type of experimental setup in Figure 3, the addition of biochar significantly increased the bacterial Shannon index in field (*p* < 0.0001) and pot experiments (*p* < 0.05) but it decreased significantly the bacterial Shannon index in laboratory setups (*p* < 0.01). There was no significant effect on soil bacterial Shannon index in greenhouse studies (*p* > 0.05).

Figure 4 summarizes that the four assessed variables of local mean annual precipitation (MAP, for field experiments only), biochar preparation temperature, type of biochar raw material and the level of biochar addition all significantly affected the bacterial Shannon index. For the 171 field studies, a mean precipitation between 200–400 mm and above 800 mm both significantly increased the bacterial Shannon index (Figure 4, *p* < 0.001 and *p* < 0.0001, respectively) whereas MAP below 200 mm or between 400 and 800 mm did not have a significant effect. The addition of biochar prepared at temperatures between 350–550 °C significantly (*p* < 0.0001) increased the bacterial Shannon index, while the addition of biochar prepared below 350 °C significantly (*p* < 0.05) decreased the bacterial Shannon index. Only biochar prepared from herbaceous raw materials significantly increased the bacterial Shannon index (*p* < 0.0001), while others biochar types did not show a significant effect on bacterial Shannon index (Figure 4). The fraction of biochar added to the soil showed significant effects on soil bacterial Shannon for both high and low levels (*p* < 0.05), but medium level additions did not show a significant effect (*p* > 0.05, Figure 4).

#### 3.1.2. Effects of Individual Parameters on the Soil Bacterial Diversity Chao1 Index

Based on 277 datasets that listed bacterial Chao1 indices, the application of biochar significantly increased this diversity index compared to the respective control groups (Figure 5) and now the effect was more strongly significant than for the reported effects on Shannon (*p* < 0.0001). However, similar as seen for the Shannon index, the overall heterogeneity was large, with *Qt* = 136,583 in the heterogeneity test, again requiring an analysis of explanatory variables.

The Orchard plot for Chao1 diversity (Figure 6) shows that among the different experimental conditions, biochar addition in pot and field experiments significantly increased bacterial diversity (*p* < 0.05 and *p* < 0.0001, respectively), similar to the observations based on the Shannon index. The reported effects on Chao1 bacterial diversity were not significant for laboratory or greenhouse experiments (*p* > 0.05).

The degree of influence of the various explanatory variables by subgroup analysis on bacterial Chao1 diversity is shown in Figure 7. In the 138 field experiments reporting Chao1 indices, biochar applied with all three analyzed levels of precipitation resulted in significantly increased bacterial diversity, with the highest significance (*p* < 0.0001) observed for MAP > 800 mm. At a biochar preparation temperature between 350–550 °C, the used biochar significantly (*p* < 0.0001) increased the bacterial Chao1 diversity, similar to the Shannon findings. Biochar prepared from herbaceous material or wood significantly increased the Chao1 index (*p* < 0.0001, and *p* < 0.001, respectively), while biochar prepared from other sources or prepared with modifications did not significantly affect the bacterial Chao1 index (Figure 7). All three application levels of high (*p* < 0.005), medium (*p* < 0.001) and low (*p* < 0.0001) amounts significantly increased the bacterial Chao1 index. The observed effects produced similar values for high (0.0643) and medium ratios of biochar (0.0688), which were both larger than the effect value for low ratios of biochar addition (0.0486). A similar trend is noted for the Shannon index shown in Figure 4.

### 3.2. Fungi Diversity

Fewer datasets were available on the effect of biochar application on fungal diversity. From the 95 papers that were analyzed, 85 datasets for fungal Shannon index and 69 datasets for fungal Chao1 index were identified. The effect of biochar addition on the fungal diversity (either Shannon or Chao1) was not significant compared to the control groups (*p* > 0.05, Figure 8). The limited number of studies that could be included here may have contributed to the non-significant results. In view of this, a subgroup analysis was not performed.

## 4. Discussion

### 4.1. Response of Shannon Index and Chao1 Index of Soil Bacteria to Biochar Addition

The purpose of this meta-analysis was to assess the effects of biochar application on soil microbial diversity, as determined by the Shannon and Chao1 index indicators. The Shannon index represents the abundance and evenness of species in a sample [82], and the Chao1 index can represent species richness and is sensitive to changes in some rare species [83]. Both the Shannon and Chao1 indices of soil bacteria were significantly higher, indicating an increase in the abundance and evenness of soil bacteria under biochar application conditions, as well as an increase in rare bacterial species. To clarify the importance of various explanatory variables on the changes in Shannon and Chao1 indices of soil bacteria after biochar application, we conducted a multifactorial importance analysis. The biochar preparation temperature and field precipitation were the most important for biochar to improve the soil bacterial Shannon index and field precipitation was the most important for the soil bacterial Chao1 index (Appendix A). Biochar preparation temperature is an important factor affecting the quality of biochar, which in turn has a very important role in improving the Shannon index of soil bacteria, and some rare bacterial species have high moisture requirements in the habitat thus precipitation is very important for the Chao1 index of soil bacteria.

The results showed that the application of biochar in soil could increase the diversity of soil bacteria at least under some conditions, and this is consistent with other meta-analyses. For example, a meta-analysis by Singh et al. also supports the conclusion that biochar can enhance soil bacterial diversity [66]; and the meta-analysis conducted by Li et al. found that biochar could significantly increase soil bacterial diversity under some conditions [84]. In this paper, the effects of biochar on the Shannon and Chao1 indices of soil bacteria were also different, for example, biochar prepared at low temperatures significantly reduced the Shannon index but had no effect on the Chao1 index, which may be due to the fact that the biochar prepared at low temperatures contained higher volatiles that were not absorbed by most bacteria but had no effect on rare bacterial species, thus causing this result; under the precipitation level of 400–800 mm, biochar significantly increased the Chao1 index of soil bacteria but did not significantly affect the Shannon index. We suggest that this level of precipitation may have stimulated the growth and reproduction of some rare soil bacterial taxa to a certain extent and thus increased the Chao1 index. For the reliability of the results, we performed a publication preference diagnosis and presented it in the form of a funnel plot (Appendix A).

The increase in soil bacterial diversity as a result of biochar application may be due to the different organic compounds contained in the biochar, while the porous structure of biochar provides a continuous supply of nutrients and expands the ecological niche for microorganisms [85]. The different effects of biochar on soil bacterial diversity under different conditions are consistent with the study of Li et al. [84]. When biochar is applied at a certain concentration, it can change the pH, nutrient concentration, water holding capacity and other physicochemical properties of the soil, which is beneficial to the growth and reproduction of soil bacteria, thus increasing their diversity [86]. The lack of an effect on the diversity of soil fungi, suggests that soil bacteria are more mobile and can enter the pores of biochar more easily, or are more adaptive and can utilize more nutrients and mineral elements, which is consistent with the results of He et al. [87]. Unlike bacteria, the soil fungal communities are generally stable and less variable [88]. The fungal mycelia can be adsorbed onto the granular structure of soil, resulting in poorer mobility compared to bacteria. A poorer absorption of nutrients and mineral elements brought by the biochar may also limit the reproductive advantage for fungi, and we believe that this may be the reason why the application of biochar does not have a significant effect on fungal diversity.

In addition, the taxonomic level (e.g., phylum or family level, etc.) on calculating soil microbial diversity may also affect the result of this study. The reason is that the numbers and community compositions of soil microorganism in different taxonomic level are different, and the α diversity indices are inextricably linked to the numbers and community compositions of soil microorganism. For example, the Chao1 index is sensitive to rare species, and the value of Chao1 index will be affected in the phylum level compared to other taxonomic levels. Similarity, the other α diversity (Shannon and Simpson index) of soil microorganism varies at different taxonomic levels. In our study, the data we collected did not consider above reason and this may cause some error on our result. So, it is necessary to consider the same taxonomic level in the future calculation and collation of soil microbial diversity. 

In our meta-analysis, only a small number of papers were located outside of China. These regions have different climates from China, where four major climates are distributed, including temperate continental, highland alpine, temperate monsoon, and subtropical monsoon climates. The climates of the study sites outside of China include semi-arid Mediterranean climate, tropical rainforest climate, etc. (Appendix A). We believe that climate differences affect the response of soil microorganisms to biochar addition. Climate is a global change factor, and the effects of global change factors on soil microbial diversity have been widely reported, for example, some anthropogenic global change factors (carbon-dioxide enrichment, global warming, Nitrogen deposition, etc.) may lead to changes in soil microbial diversity [89,90]. In future studies, we will also focus more on these issues from a global perspective.

### 4.2. The Response of Soil Bacteria to Biochar Addition for Different Experimental Types

The total effect value reported here through the random effects model showed that biochar could significantly improve soil bacterial diversity, however, the Q-test identified a high overall heterogeneity, necessitating the use of a mixed effect model to evaluate the degree of influence of different experimental types and subgroup variants.

The reported diversity indices for soil bacterial communities were generally higher for greenhouse, pot and laboratory experiments than for field experiments. However, in our meta-analysis, the effect of biochar on enhancing soil bacterial diversity was higher in the field experiments than in other experimental setups. We believe that there are two reasons for this: i. The field experiments produced the largest dataset of all experimental types, and the larger the dataset, the more reliable the outcome is. ii. In most laboratory, greenhouse and pot experiments the soil bacterial habitat would be more stable, without dramatic fluctuation in diurnal or daily temperature changes, and with an adequate and constant nutrients supply there would be fewer harsh environmental conditions, thus leading to an insignificant biochar response. In contrast, in the field experiments, the habitat conditions of soil bacteria would be more strongly fluctuating, and when biochar is present this amendment can obviously enhance soil quality, with soil bacteria responding more to it, as shown by an increase in diversity.

### 4.3. Effect of Biochar Addition Level on Soil Microbial Diversity

The meta-analysis identified that the response of soil bacterial diversity depended on the addition level of biochar. At a high level of addition (>3%) biochar significantly increased the Shannon index of soil bacteria (Figure 4) and both high and medium (1.5–3%) levels of biochar significantly increased the Chao1 index of soil bacteria (Figure 7).

Biochar is a carbon-rich material, and carbon is one of the nutrients necessary for biological growth and reproduction. We consider an increased metabolic capacity and activity of soil bacteria is likely after the addition of biochar, and the large surface area and porous structure of biochar can provide space for bacteria to live, thus leading to an increase in bacterial diversity in the soil after the addition of biochar. When the amount of biochar added to the soil increases, the soil bacteria have more habitats and are less likely to be preyed on, thus increasing the diversity of soil bacteria [91]. Of course, the effect of biochar application on soil bacterial diversity is different in different studies, for example, Masiello et al. suggest that high application rates of biochar can destroy the microenvironment for microbial growth and thus reduce soil bacterial diversity [92]; Gomez et al. suggest that high application rates of biochar can introduce toxic components and cause a decrease in microbial diversity [93]; in addition, Ameloot et al. suggest that high application rates of biochar can limit microbial carbon metabolism and thus reduce microbial diversity [94].

### 4.4. Biochar Application and Precipitation

Precipitation levels in the locations of field experiments was selected as one of the explanatory variables in an attempt to explain the large heterogeneity of the overall effect value, for which the data were grouped in arid (<200 mm), semi-arid (200–400 mm), semi-humid (400–800 mm) and humid conditions (>800 mm). The subsequent subgroup analysis (Figure 4 and Figure 7) showed that for both bacterial diversity indices, application of biochar under arid conditions was not beneficial. The strongest beneficial effect was observed for humid conditions (*p* < 0.0001). Precipitation levels affect the soil water content, which as an important indicator of soil quality and is the main influencing factor of soil microbial diversity [95]. We assumed that precipitation plays an important role on soil bacterial diversity under biochar addition, as a higher soil water content might increase the mobility of soil bacteria, which largely improves their survival ability; in addition, soil water liberates nutrients and mineral elements from the biochar and enables adsorption by soil bacteria resulting in a larger nutrient enrichment zone around the biochar in humid conditions. Thus, precipitation becomes an important influencing factor of soil bacterial diversity by changing the soil water content, which is consistent with the findings of many studies. For example, Yang et al. found that a certain amount of precipitation can drive higher diversity of soil bacteria [96]; Bickel et al. found that soil bacterial diversity was highest at moderate soil water content; Tu et al. revealed that soil microbial diversity in China’s temperate steppe was significantly correlated with average annual precipitation [97]. Whether precipitation has a significant effect on soil bacterial diversity after addition of biochar will further be affected by different climatic conditions, altitudes, and soil bacterial habitat types.

### 4.5. Different Biochar Preparation Materials and Temperature

Since biochar can support growth of soil microorganisms by the release of nutrients in the soil, we analyzed what role of raw materials and preparation temperatures of the applied biochar had. For this, we used the Q-test. The *Qm* value represents the heterogeneity caused by the explanatory variables, with a higher value indicating a greater effect of the explanatory variables on the reported diversity. Based on the Shannon index, the different raw materials for biochar preparation produced a *Qm* value of 18.3591 (*p* < 0.05) and the different preparation temperatures resulted in a *Qm* of 21.0141 (*p* < 0.05). This illustrated that both the biochar type and the preparation temperature had a significant modulating effect on soil bacterial diversity by Shannon (*p* < 0.01) (Figure 4). Biochar with different preparation materials (*Qm* = 50.3364) and temperatures (*Qm* = 36.8704) also has a significant regulatory effect on the Chao1 index of soil bacteria (*p* < 0.0001). (Figure 7). Among the different biochar preparation materials, herbaceous sources most strongly increased soil bacterial diversity (*p* < 0.0001 for both Shannon and Chao1). The properties of biochar prepared from different raw materials will vary considerably. For example, Lu et al. found that biochar produced from herbal materials had larger pore sizes and larger transmissive pore volumes [98]. Brendova et al. reported that biomass and lignin content affected the pore structure of biochar, and that biochar prepared from herbaceous materials had a higher skeletal density [99]. Soria et al. considered biochar prepared from herbaceous materials more suitable for improving and repairing heavy metal contaminated soil than biochar prepared from other materials [100].

As for the different biochar preparation temperatures, pyrolysis of raw materials at 350–550 °C most strongly increased soil bacterial diversity (*p* < 0.0001 for both indices). That the pyrolysis temperature affects biochar properties was demonstrated in several studies, for example, Das et al. found that the cation exchange capacity of biochar increased with increasing pyrolysis temperature, but the organic carbon content decreased [101]. We consider this as a possible explanation why the performance of biochar prepared at medium pyrolysis temperature may result in higher microbial diversity. Han et al. applied six types of biochar produced with various pyrolysis temperatures for soil improvement-related experiments, and found that biochar resulting from 400 °C pyrolysis had the greatest capacity for soil *p* adsorption [102]. Lastly, Xu et al. found that the reduction capacity of biochar prepared within relatively moderate pyrolysis temperature range increased with increasing temperature, but when produced at high temperatures the reduction capacity of the biochar decreased with increasing temperature [103].

## 5. Conclusions

Our meta-analysis was performed to address two scientific questions: Does the addition of biochar to soil affect soil microbial diversity? If so, which factors mostly affect this? After analysis of the data extracted from 95 publications using a random effects model and a mixed-effects model we can draw the following conclusions: (i) the addition of biochar to soil can have a significant effect on soil bacterial diversity, though not on soil fungal diversity; (ii) the effect of biochar on soil bacterial diversity was significantly influenced by the type of raw material used for biochar preparation, by the temperatures of biochar pyrolysis and by the addition ratios of biochar to soil; (iii) field experiments were most often recorded in the set of the analyzed data, and these may be the most suitable to assess the effect of biochar on soil bacterial diversity under natural conditions. In our meta-analysis, the field experiments resulted in a greater overall regulatory effect compared to other experiment types; (iv) for field experiments, the increase of biochar-dependent soil bacterial diversity was greater under medium precipitation conditions; (v) medium level biochar addition is more effective in enhancing soil bacterial diversity than low or high level addition; (vi) biochar prepared from herbaceous materials can better enhance soil bacterial diversity than biochar produced from other sources; (vii) biochar prepared at moderate pyrolysis temperatures best promotes soil bacterial diversity. Therefore, we believe that it is crucial to optimize the quality of biochar before application in future research on the application of biochar in soil. This includes the selection of suitable raw materials for preparation, the appropriate preparation temperature and the amount of biochar to be applied. Finally, by optimizing biochar addition measures and adapting biochar addition programs in different regions, and promote the advantages of soil microorganisms, to improve the soil quality and sustainable in soil utilization in the further.

## Figures and Tables

**Figure 1 microorganisms-11-00641-f001:**
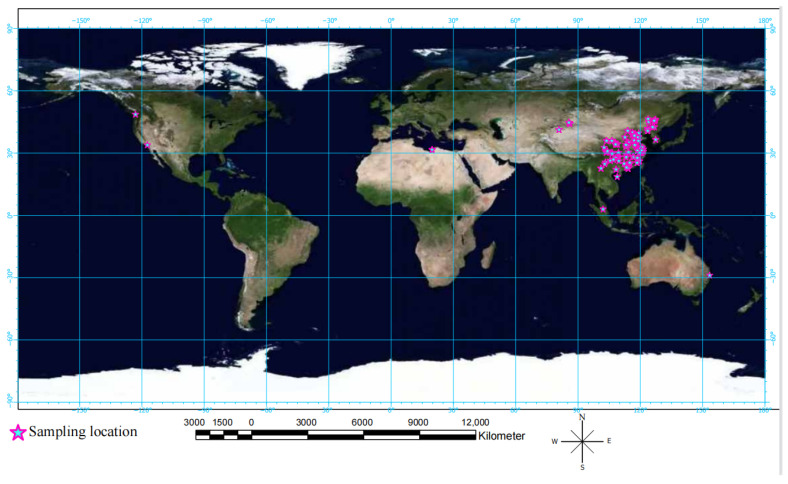
Global distribution of the study sites of the literature used in the meta-analysis. Only seven studies were located outside China. The sampling points were plotted with “Arcgis Pro2.5” software.

**Figure 2 microorganisms-11-00641-f002:**
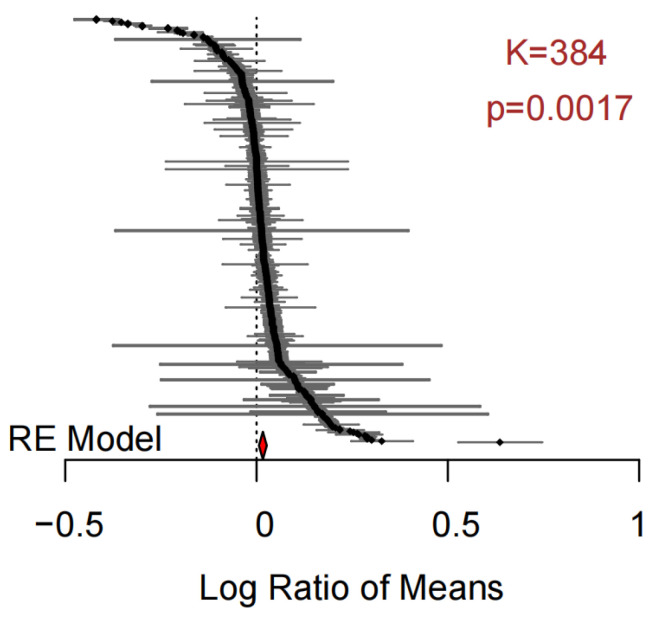
Forest plot collating the effect size estimates for biochar addition on bacterial Shannon index obtained from 384 datasets. Log response ratios (effect sizes) are shown as black dots with 95% confidence intervals (Cl_95%_) as gray lines. The overall weighted mean effect size estimate is shown as a red diamond at the bottom. The dashed gray line indicates a Log response ratio of zero when the RE model would imply a random effect.

**Figure 3 microorganisms-11-00641-f003:**
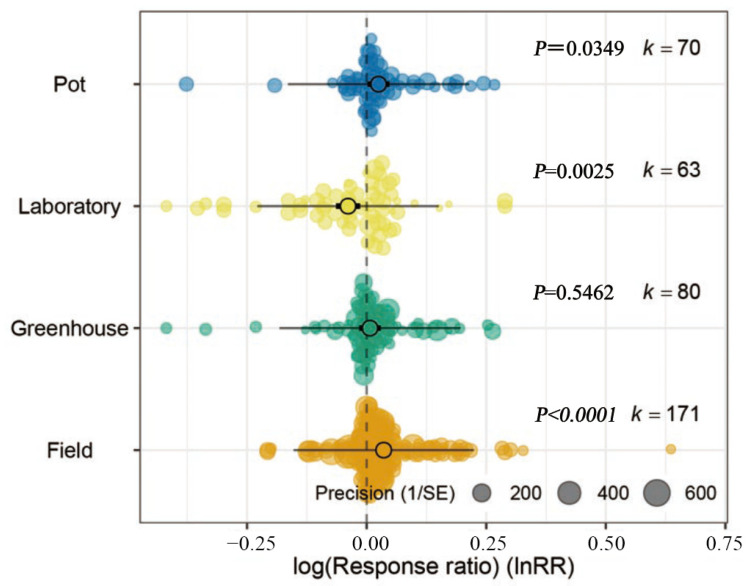
Orchard plot showing mean estimates (open circles) of bacterial Shannon index following biochar application, with individual effect sizes (colored circles) for four types of experimental setups. The precision (inverse variance) is plotted with Cl_95%_ as bold lines and the prediction interval as gray lines. The number of studies is given as k.

**Figure 4 microorganisms-11-00641-f004:**
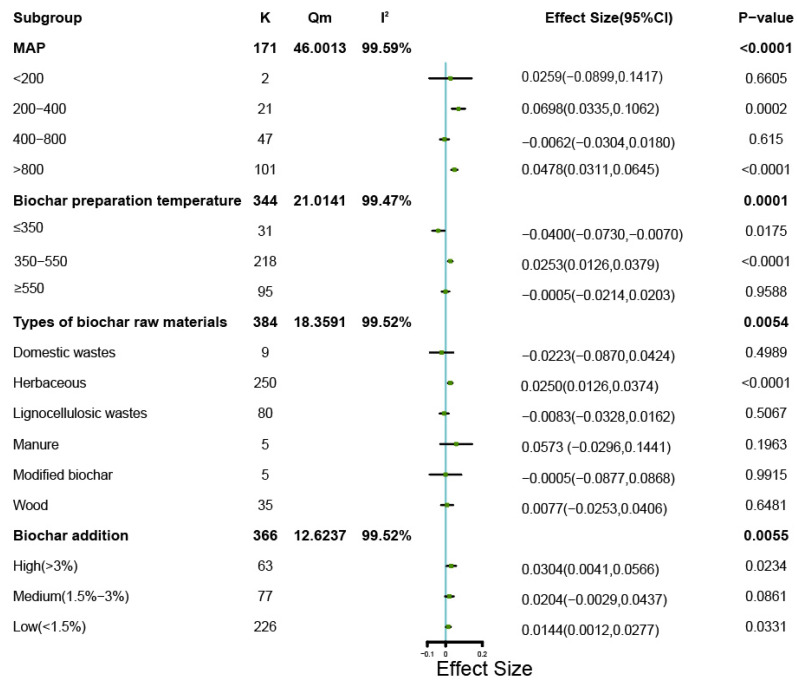
Forest plot showing mean changes in bacterial Shannon index for different subgroups. The datapoints show treatment effects for a given group with bars representing CI_95%_ for the indicated categories. K specifies the total numbers of replicates from the combined studies, I^2^ is an indicator of the magnitude of inter-case heterogeneity in the response and *Qm* is the basis for determining the degree of influence of the effect of the explanatory variables on the effect value.

**Figure 5 microorganisms-11-00641-f005:**
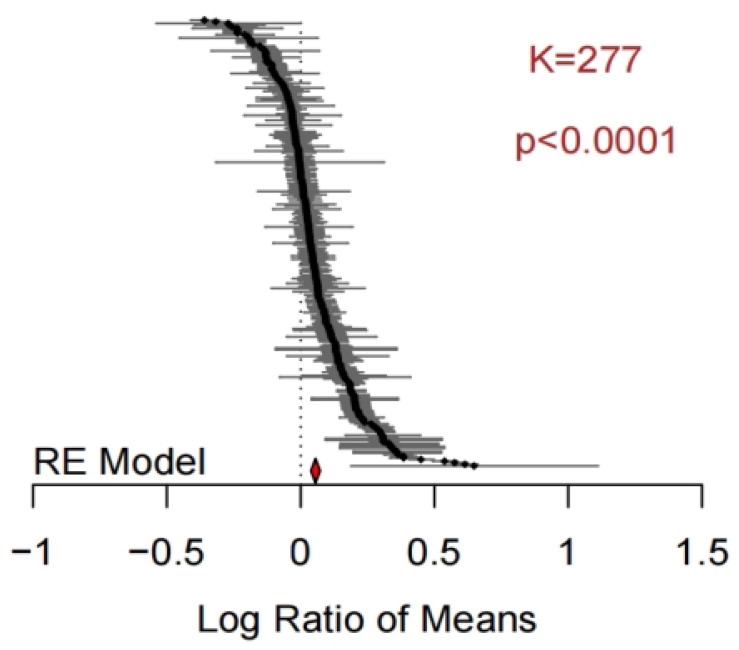
Forest plots of 277 effect size estimates for biochar addition on the reported bacterial Chao1 index. The layout of the figure is the same as in Figure 2.

**Figure 6 microorganisms-11-00641-f006:**
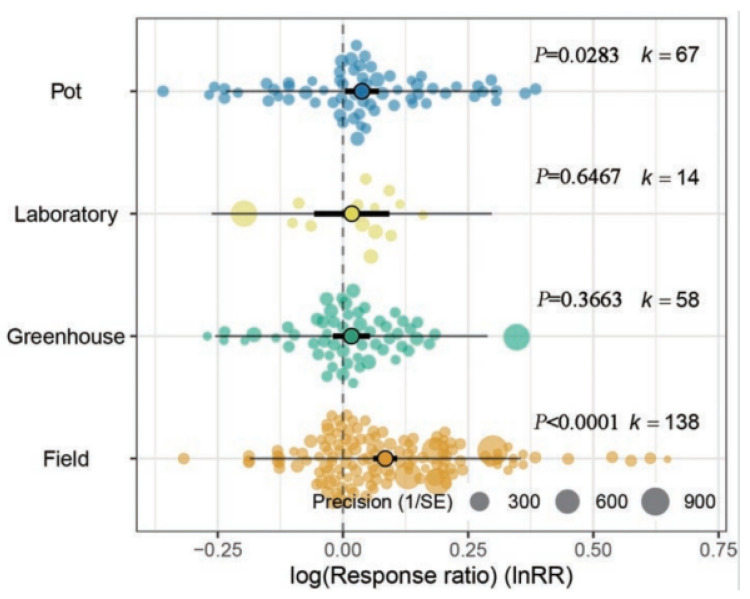
Orchard plot showing mean estimates (open circles) of bacterial Chao1 index, with confidence intervals (bold line), prediction intervals (fine line) and individual effect sizes and their precision (inverse variance) shown for different types of experimental setups. The number of studies is given as k.

**Figure 7 microorganisms-11-00641-f007:**
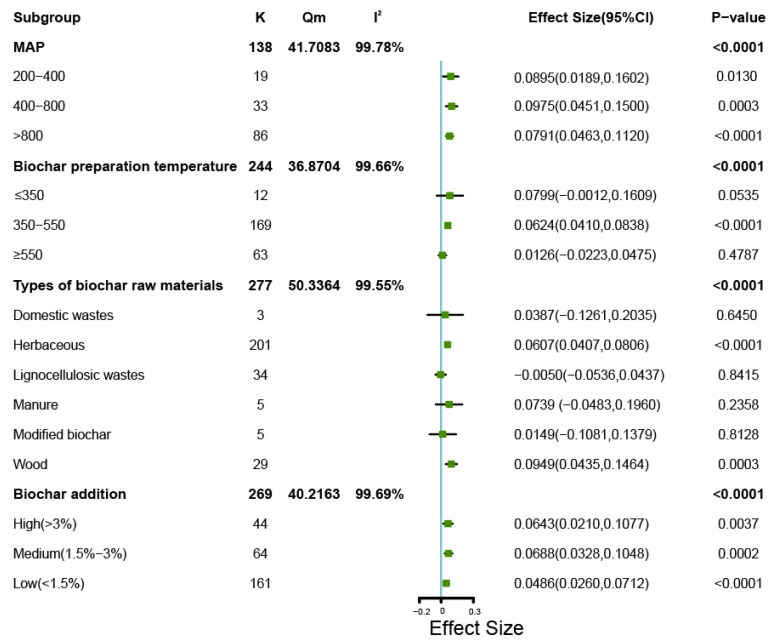
Forest plots showing mean changes in bacterial Chao1 index for different subgroups. For further explanation, see the legend of Figure 4. In the soil bacterial Chao1 index dataset, there are no areas with precipitation less than 200 mm, and therefore not shown in the figure.

**Figure 8 microorganisms-11-00641-f008:**
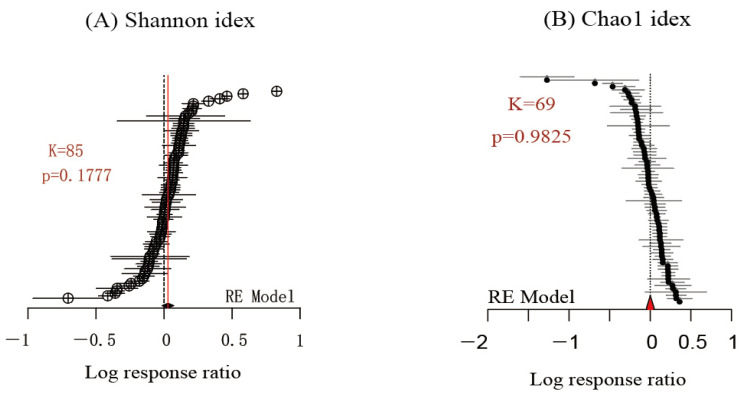
Forest plots of 85 effect size estimates for biochar addition on fungal Shannon index (**A**) and of 69 data sets on fungal Chao1 index (**B**).

## Data Availability

Data are contained within the article.

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
