# Peer review of "Meta-Analysis of the Effects of Biochar Application on the Diversity of Soil Bacteria and Fungi"

_microorganisms, 2023, doi:10.3390/microorganisms11030641_

Round 1
Reviewer 1 Report
The science presented is original with plenty of detail. In general, the authors have done a good job explaining the background information necessary to appreciate the rationale and results of the experiments. The manuscript was prepared correctly. Methodology and analysis of results rather don't raise any objections.
Reviewer 2 Report
The paper referred to the effect on diversity. Reconsider the title of the paper, for example “Meta-analysis of the effects of biochar application on the diversity of soil bacteria and fungi”
I expected information or a section that presumes or provides the Families, or Phyla while discussing the diversity of both bacteria and fungi. It is necessary to add this information.
Just 10% of other study locations are found outside of China. Given that the climate, for instance, differs from that of China, how significantly do these places impact the indices? Add this information.
There are further typographical errors, therefore the authors need to pay attention to the punctuation, spaces, and parentheses.
Reviewer 3 Report
The results were very interesting and corroborated with the literature. However, the authors left something to be desired in the discussion of the results, they could have explored the discussion further, emphasizing the importance of this study;
Work more on the discussion.
Reviewer 4 Report
Manuscript ID: microorganisms-2238083
Title: Meta-analysis of the responses of soil bacterial and fungal diversity to biochar applications
In the manuscript, a meta-analysis of information related to soil biochar addition effects on microbial diversity (Shannon and Chao1 indexes). Resulting of the analysis, key factors that influence increases in bacterial such as the type of raw material used for biochar preparation, the temperatures of biochar pyrolysis, and the addition ratios of biochar to soil, were identified, while respect to the fungal diversity, this is not strongly influenced by the biochar addition. The conducted analysis is adequate, and the result can improve the actual soil biochar addition approaches, but this in not discussed or highlighted in the conclusions.
Please address the following commentaries:
General commentaries:
Review in the whole manuscript if the format for introducing the references will be “conditions [1-3]” or “conditions[1-3]”, eliminate extra space, or add missing space according to the adequate format.
The format of the information in figures 4 and 7 must be carefully reviewed, there are different spaces between words/information are missing.
Review all temperature and precipitation values in the whole manuscript, there are presented in different formats, choose adequate format and homogenize.
In conclusion, information about how the findings of the present study could be applied to enhance the beneficial effects of biochar addition, or reduce adverse effects, to maximize beneficial effects of biochar addition on microbial diversity, which characteristics or key parameters have to be taken in account in biochar production.
Additional commentaries:
Line 12, add a space between the words “Sciences,Harbin”
Line 22, eliminate extra space between words “soils,of”
Line 40, eliminate extra space between words “biomass as”
Line 44, eliminate extra space between words “the microbial”
Line 60, eliminate extra space in “soil ,”
Line 66, eliminate extra space between words “rhizosphere where”
Line 72, in CO2, the number must be presented as subscript.
Line 78, eliminate extra space between words “studies found”
Line 85, eliminate extra space between words “soils,of”
Line 86, add a space between the words “dominated.These”
Line 130, eliminate the first % symbol in “(1.5%-3%)”
Line 151, (95%CI) could be (95% CI)”
Line 154, 95%CL is 95%CI, or is a different parameter?
Line 156, correct the following “tively .If”
Line 156, review, decide which will be the format for 95%CI and homogenize in the whole manuscript, there is presented in many different forms
Line 159 and 160 P value is presented in two formats (P< 0.05 ) (P < 0.05 ), in two for there are extra space between the number and the parenthesis close.
Line 160, eliminate extra space between “(P < 0.05 ) .”
Line 174, correct the format in “were summed. .a significant”
Line 184, correct the format in “present [76].To test”
Line 185, eliminate extra space in “Egger’s test [77] ,”
Lines 185 and 186, correct the format in “The“Trim and fill”algorithm”
Line 187, eliminate extra space in “error [79] .”
Line 203, eliminate extra comma in “(Fig. 2, P<0.05),.The”
Line 236, “Figure 4” could be “Fig. 4”
Line 240, use superscript for number in “I2”
Line 245, correct the format in “group s(Fig. 5)”
Line 255, eliminate extra space in “diversity (P<0.05 and P<0.0001…”
Line 279, correct the format in “of Fig. 4.In the”
Line 300, correct the format in “bacterial species.To”
Line 303, correct the format in “ysis.The”
Line 313, correct the format in “diversity[66];and”
Line 315, correct the format in “tions[84].In”
Line 320, correct the format in “result;under”
Line 324, correct the format in “index.For”
Line 329, correct the format in “microorganisms[85].The”
Line 370, review redaction in “…of biochar When the amount…”
Line 372, review redaction in “soil bacteria [89] Of course”
Line 410, eliminate extra period in “(P<0.05)..”
Line 413, correct the format in “preparation materials(Qm=50.3364) and temperatures(Qm=36.8704) also”
Line 414, correct the format in “bacteria (P<0.0001).(Fig. 7).”
